# Traumatic Brain Injury Induces a Differential Immune Response in Polytrauma Patients; Prospective Analysis of CD69 Expression on T Cells and Platelet Expansion

**DOI:** 10.3390/jcm11185315

**Published:** 2022-09-09

**Authors:** Alexander Ditsch, Lea Hunold, Friederike Hefele, Frederik Greve, Olivia Mair, Peter Biberthaler, Laura Heimann, Marc Hanschen

**Affiliations:** 1Experimental Trauma Surgery, Klinikum Rechts der Isar, School of Medicine, Technical University of Munich, Ismaninger Str. 22, 81675 Munich, Germany; 2Department of Trauma Surgery, Klinikum Rechts der Isar, School of Medicine, Technical University of Munich, Ismaninger Str. 22, 81675 Munich, Germany

**Keywords:** traumatic brain injury, neuro-immune interaction, T helper 17 cells, CD4+ regulatory T cells, CD69, trauma, hemostasis

## Abstract

Background: Accidents and injuries are the leading causes of mortality in young people. CD4+ regulatory T cells (CD4+ Tregs), Th17 cells and platelets could be identified as key players in post-traumatic immunological dysfunction, which is a common cause of late mortality in trauma patients. The mechanisms of activation of these cell types and their interaction remain mostly unclear. Since CD69 is not only a leukocyte marker but has also immunoregulatory functions, we postulate a role for CD69 after trauma. The present study investigates the expression of CD69 on CD4+ Tregs and Th17 cells, as well as the posttraumatic expansion of platelets and hemostatic function. Subgroup analysis was performed to assess the differences between polytrauma patients with and without severe traumatic brain injury (TBI). Methods: In this non-interventional prospective clinical trial, we analyzed sequential blood samples over a period of 10 days from 30 patients after multiple traumas with an ISS ≥ 16. Platelet function was assessed by rotational thromboelastometry (ROTEM analysis). CD4+ Tregs and Th17 cells were stained with surface markers and analyzed by flow cytometry. Results: We were able to demonstrate a significantly increased expression of CD69 on CD4+ Tregs after trauma. Subgroup analysis revealed that the absence of severe TBI is associated with a significantly higher expression of CD69 on CD4+ Tregs and on Th17 cells. Platelets expanded and showed signs of dysfunction, while an overall tendency of posttraumatic hypercoagulation was detected. Conclusions: Our results support the concept of injury-specific immune responses and add to a further understanding of the complex pathophysiology of post-traumatic immune dysfunction.

## 1. Introduction

Trauma is the leading cause of death in people under the age of 44 years [1]. Only a part of this mortality due to accidents and injuries is caused directly by trauma. An important risk factor for the high mortality rate after trauma is the development of the posttraumatic systemic inflammatory response syndrome-SIRS [2]. Up to 90 percent of patients with multiple traumas develop SIRS in the first week after trauma [3]. SIRS varies with the extent of the trauma and can lead to organ damage or even multi-organ dysfunction syndrome (MODS) [4]. MODS is a serious condition requiring intensive care treatment, associated with mortality of 30 to 80 percent [5]. SIRS is often accompanied by compensatory anti-inflammatory response syndrome (CARS). Recent evidence suggests that pro- and anti-inflammatory responses occur in parallel [6]. Usually, the positive effects of the involved mediators predominate. An imbalance between these forces is harmful and can lead to shock, barrier and coagulation disorders, as well as anergy and immunosuppression. CD4+ regulatory T cells (CD4+ Tregs), Th17 cells and platelets could be identified as key players in post-traumatic immunological dysfunction. Since the exact mechanisms of immune regulation after multiple trauma are still poorly understood, further basic research is urgently required in order to improve diagnostics and therapy for seriously injured patients in the future.

CD4+ regulatory T cells (CD4+ Tregs) play an important role in the regulation of immune function and MODS. CD4+ Tregs represent a subgroup of T cells exerting anti-inflammatory effects on the organism and contributing to the self-tolerance of the immune system. They were first characterized by the expression of CD25 and account for approximately 10% of CD4+ T cells [7]. CD4+ Tregs develop a higher TCR diversity than effector T cells, a large repertoire of TCR is needed to ensure optimal immune regulation by CD4+ Tregs [8]. TNF is of particular importance in the regulation of the immune system, as it inhibits the activity of CD4+ Tregs and thus influences their immunosuppressive function [9]. The transcription factor forkhead box P3 (FoxP3) plays a major role in the development and function of CD4+ Tregs [10]; thus, it is often used as a marker for their identification. As FoxP3 has been shown to be non-specific for regulatory T cells, recent findings suggest that the specificity and sensitivity of CD4+ CD127 low cells are comparable to that of CD4+ FoxP3+ cells, and can be used as an alternative to ensure the identification of CD4+ Tregs [11]. Different subgroups of CD4+ Tregs have been shown, but their exact roles are not yet clear and therefore they are the subject of current research. The benefit of CD4+ Tregs on the outcome after SIRS and sepsis is not fully understood. In a sepsis mouse model, it has been shown that the transfer of in vitro stimulated CD4+ Tregs was protective [12]. Other studies have shown no benefit for overall survival in the sepsis model, despite increases in cell numbers and enhancement of immunosuppressive function [13]. Recent studies conducted by our group described the kinetics and localization of CD4+ Treg activation following trauma in a murine model [14] and found that injury induces rapid activation of CD4+ Tregs but not CD4+ non-Tregs. Furthermore, most recently we were able to describe an increase in IL-17A expression on CD4+ Tregs in patients following multiple trauma [15].

Th17 cells are a subtype of T helper cells involved in the defense against bacteria and fungi and play a role in the pathogenesis of autoimmune diseases. The surface expression of IL-17A is a specific marker of Th17 cells [16]. They are the major source of IL-17 and can cause the activation of CD8+ T cells by its secretion [17]. Other secretory products are IL-17F, IL-21, IL-22, IL-6 and TNF-α. Receptors for these cytokines are found on a variety of immune cells, including neutrophils, lymphocytes, epithelial, endothelial, and fibroblast cells. These secreted messenger substances lead to inflammatory processes in the target cells and to the infiltration of further immune cells. CD4+ Tregs seem to play a role in the regulation of Th17 cells. These two subtypes influence each other, and the ratio of Th17 cells to CD4+ Tregs seems to have a significant influence on the development and outcome of inflammatory and autoimmune diseases [18].

Platelets have long been known for their contribution to coagulation, but they also play a role in the processes of the innate and acquired immune response [19]. By endothelial injury, subendothelial structures are exposed and platelets are activated by collagen, von Willebrand factor (vWF), thrombin, ADP and other factors. Platelets are involved in inflammatory processes, interact with bacteria, recognize bacterial peptidoglycans and lipopolysaccharides via toll-like receptors such as TLR2 and TLR4 and enhance immune responses [20]. An overview of platelet-mediated inflammatory mediators is provided by Klinger and Jelkmann [21]. The influence of platelets on the outcome of SIRS and sepsis is contradictory in the literature. Not only do they improve antigen presentation and CD8+ T cell responses, but they also have implications for T cell-dependent humoral immunity [22]. Platelets also specifically regulate the immune responses of Th1, Th17 and regulatory T cells, promoting regulatory T cells but appearing to have a biphasic effect on the activation of Th1/Th17 cells [23]. Furthermore, they significantly influence coagulation after severe trauma [24]. The incidence of coagulation disorders after trauma is high and is considered an independent predictor of mortality [25]. Even mildly diminished platelet activation seems to be a sign of coagulopathy associated with increased mortality in trauma patients [26].

The interaction between CD4+ Tregs, Th17 cells and platelets has been investigated several times in animal models already. We could previously show reciprocal activation of CD4+ Tregs and platelets as part of the protective immune response following trauma-induced injury in a mouse model [27]. Post-traumatic interaction seems to function, at least in part, via paracrine interactions through TLR4 and TNFR2-dependent pathways [28]; furthermore, GPIIb/IIIa-, PAR4- and fibrinogen-dependent pathways in platelets have most recently been shown to modulate CD4+ Treg baseline activity [29].

Traumatic brain injury (TBI) is one of the most common injuries in severely injured patients, therefore a subgroup analysis was performed in this study to assess the effect of TBI on posttraumatic immune disturbance. Our current understanding is that damage is first due to the trauma itself and secondary through hypotension and hypoperfusion of the tissue. Recent findings underline that, following the injury of the brain, one of the defense mechanisms of the innate immune system is the development of neuroinflammation. Microglia plays an important role in this complex system. It is able to produce neuroprotective factors and induce neurological recovery, as well as pro-inflammatory and cytotoxic mediators, which can lead to neuronal dysfunction and cell death [30]. In this context, little is known about the role of CD4+ Tregs and the adaptive immune system following TBI in trauma patients. Hemostasis and coagulopathy are highly related to the outcome of TBI [31]. Patients with impaired hemostasis have a high risk for intracranial bleeding and further injury to the brain. Further impairment is the result of a variety of different factors like the distribution of inflammatory, procoagulant or anticoagulant mediators, fibrinolysis and the disrupted blood-brain barrier. Traumatic coagulopathy is often associated with extracranial injuries like substantial blood loss (hemorrhagic shock), consumption, hypothermia or acidosis, whereas the pathogenesis of TBI-induced coagulopathy remains still mostly unclear [32]. In this context, little is known about the immunologic role of platelets following TBI in trauma patients.

Taken together, this prospective non-interventional clinical trial aims to improve the understanding of post-traumatic interactions between CD4+ Tregs and platelets in humans, to ensure better care for multiple trauma patients in the future. Due to the fact that TBI constitutes one of the most common injuries in severely injured patients, a subgroup analysis was conducted to investigate the influence of severe TBI.

## 2. Material and Methods

### 2.1. Patients

The study was conducted at the Department of Trauma Surgery of the University Hospital Rechts der Isar in Munich. Thirty patients were included in this prospective study between December 2014 and December 2018. Patients were eligible for inclusion in the study when brought to the emergency no more than 12 h after their respective trauma. Furthermore, the inclusion of the patient depended on the availability of the study team for the individual follow-up period. Written informed consent was obtained from all patients included or their relatives according to the patient’s suspected will. The study follows the principles of the Declaration of Helsinki with its novelizations of Tokyo 1975, Hongkong 1989 and Somerset West 1996. The protocol was approved by the ethics committee of the hospital (trial number of the ethics application: 5925/13).

We recruited severely injured patients from the ages of 18 to 95 years with multiple traumata defined by an Injury Severity Score (ISS) of at least 16; isolated injuries were excluded. Pregnant women and prisoners were excluded.

### 2.2. Reagents

The staining of T-cells was performed in PBA, consisting of PBS, albumin from bovine serum and sodium azide (Sigma-Aldrich, St. Louis, MO, USA). We used Fc-block (BioLegend, San Diego, CA, USA) to prevent the non-specific binding of the antibodies used for cell staining. Surface staining was performed using anti-CD4 (OKT4) (eBioscience, San Diego, CA, USA), anti-CD161 (HP-3G10) (eBioscience, San Diego, CA, USA), anti-CD196 (R6H1) [33], anti-CD25 (BC96) (eBioscience, San Diego, CA, USA) and anti-CD127 (eBioRDR5) (eBioscience, San Diego, CA, USA). Cell staining for detection of T-cell activation was performed with anti-CD69 (FN50) (eBioscience, San Diego, CA, USA). For thromboelastometry we used extem, intem and fibtem reagents from TEM International GmbH (Munich, Germany).

### 2.3. Blood Samples and Data Retrieval

Whole blood was taken at nine different time points: immediately after the arrival in the trauma bay (TB), after 6, 12, 24, 48 and 72 h, as well as after 5, 7 and 10 days (The mean time from injury to admission in the trauma bay was 60.5 min). The blood was collected into monovettes (Sarstedt AG & Co., Nümbrecht, Germany) containing citrate for thromboelastometry and ethylenediaminetetraacetic acid (EDTA) for cell staining and flow cytometry. In addition to the series of nine blood draws, clinical patient data were collected (demographics, injury pattern, injury severity and outcomes) and platelet count (routine blood work) was noted.

### 2.4. Flow Cytometry

The detection and differentiation of T-cell subpopulations (Th17 cells and CD4+ Tregs), as well as the analysis of surface activation markers, were performed by flow cytometry on a MACSQuant^®^ 9 from Miltenyi Biotec [33].

First, EDTA blood was transferred to a tube with Schwinzer solution, consisting of aqua with ammonium chloride (Carl Roth GmbH, Karlsruhe, Germany), potassium carbonate (Caesar & Lorentz GmbH, Hilden, Germany) and EDTA (Carl Roth GmbH, Karlsruhe, Germany), for 15 min at 4 °C for hemolysis of erythrocytes. The cell solution was then washed and buffered with 210 µL PBA, which consists of PBS, albumin from bovine serum and sodium azide (Sigma-Aldrich, St. Louis, MO, USA). Following Fc-block, specific surface antibodies were added for the staining of Th17 and CD4+ Treg subpopulation. For Th17 T-cells, we used anti-CD4 (APC-labelled), anti-CD161 (ef450-labelled) and anti-CD196 (FITC-labelled). CD4+ Tregs were stained with anti-CD4 (AlexaFlour^®^ 488-labelled), anti-CD25 (ef450-labelled) and anti-CD127 (APC-Cy7-labelled). For the detection of activation, we used surface staining antibodies anti-CD69 (PE labelled). After incubation at 4 °C for 15 min, the cells were washed, resuspended with PBA and immediately analyzed. According to the manufacturer’s recommendation, we used calibration beads from Miltenyi Biotec [33] to calibrate the MACSQuant^®^ before each measurement. Appropriate instrument settings and compensation were applied prior to running the samples to minimize variability between experiments. Regular bleaching and flushing of the instrument were performed to ensure clean tubes and flow chamber.

Data analysis was performed using FlowJo Software v.10 (FlowJo LLC, Ashland, OR, USA). Single cells were selected by gating out doublets and cell debris. We identified CD4+, CD25+ and CD127- cells as CD4+ Tregs, and CD4+, CD161+ and CD196+ cells as Th17 cells, and determined the relative mean fluorescence intensity (MFI) of the cells stained with PE-conjugated anti-CD69 antibodies. The relative MFI is determined by dividing the MFI of CD4+ Treg cells stained with PE-conjugated anti-CD69 antibodies by the MFI of Treg cells not stained with any PE-conjugated antibody. CD69 expression on Th17 cells was assessed analogously.

### 2.5. Thromboelastometry

Thromboelastometry analysis was used to measure hemostatic function and platelet activation. ROTEM^®^ is an easy point-of-care device for the measurement of plasmatic and cellular coagulation processes. The clotting process is visualized and multiple clotting parameters are provided. The measurement was performed with citrated whole blood at 37 °C within 10 min of the blood collection. We used a ROTEM^®^ delta from TEM International GmbH (Munich, Germany). We used extem, intem and fibtem reagents with the automatic pipetting system to test the extrinsic and intrinsic coagulation pathway, as well as the contribution of fibrinogen to the clot. Clotting time (CT), clot formation time (CFT) and maximum clot firmness (MCF) were compared for each extrinsic, intrinsic and fibrinogen thromboelastometry at every time point. The Clotting time (CT) is the time from the beginning of the measurement until the appearance of a clot. This covers the coagulation activation, thrombin formation and onset of the clot polymerization. The clot formation time starts from the onset of the clot formation and ends when the amplitude has reached 20 mm. This matches with the fibrin polymerization and solidification of the clot by thrombocytes and factor XIII. The maximum clot firmness reflects maximum mechanical manifestation and corresponds to the strength of the clot. The clot is solidified by polymerized fibrin, platelets and factor XIII. For the evaluation of the platelet contribution of the maximum clot firmness, the surrogate marker platelet MCF was calculated as the MCF (extem-fibtem) by subtracting the MCF measured in fibtem from the one measured in extem, as has been described in literature before [34].

### 2.6. Subgroup Analysis for Patients with Traumatic Brain Injury

Subgroup analysis was performed in dependence on the existence of severe injury to the brain. To objectify this injury we determined the Abbreviated Injury Severity (AIS) score of the head region for all 30 patients. We declared the absence of severe traumatic brain injury by an AIS of 1-2 and the existence of severe traumatic brain injury by an AIS ≥ 3.

### 2.7. Statistics

Data were analyzed using IBM^®^ SPSS-Statistics Software. The values of the descriptive statistical analysis are presented in the form of mean ± standard deviation. Generalized estimating equations (GEEs) with an exchangeable correlation matrix were applied to test for statistically significant changes over time, measured in hours. Regression coefficient (B), standard error (SE), 95% confidence interval (CI) as well as a probability value (*p*) and correlation coefficient (r) are given in brackets. *p* < 0.05 was considered significant.

## 3. Results

### 3.1. Study Population

Thirty patients were included in the study. The mean age of the patients was 50.5 ± 18.5 years. Of the patients, 60% were male and 40% were female. The average ISS was 33.2 ± 11.5. 26 patients (87%) survived the first 10 days after trauma. They spent an average of 13.3 ± 15.3 days in the intensive care unit and had an average of 3.9 ± 2.4 surgical procedures. The most injured regions were—in descending order: limb/pelvic girdle (90%), chest (80%), head/neck (67%), external (63%), face (50%) and abdomen/pelvic content (37%). Twenty patients suffered a traffic accident (66.7%), and five patients (16.7%) suffered a suspected suicide attempt.

Regarding the subgroup analysis of patients with and without severe traumatic brain injury, the non-severe TBI subgroup contained 17 patients, five women and 12 men with an average age of 47 ± 17 years. The median ISS of this subgroup was 30 ± 11 with a survival of 100%. Compared to this group we detected four females and nine male patients with severe damage to the brain with a survival of 69% in the first 10 days after trauma. The average age of the severe TBI group was 47 ± 22 years with a slightly higher median ISS Score of 37 ± 12. An overview of the entire demographic evaluation of patient data is given in Table 1.

### 3.2. Late Increase in CD69 Expression on CD4+ Tregs

A significant increase in CD69 expression on CD4+ Tregs (B = 0.033, SE = 0.0087, 95% CI [0.016, 0.050] could be detected, *p* < 0.001), illustrated in Figure 1A, while there was no significant increase on Th17 cells (B = 0.058, SE = 0.0347, 95% CI [−0.010, −0.0126, *p* = 0.093), illustrated in Figure 1B.

### 3.3. Increase in CD69 Expression in Patients without Severe TBI on CD4+ Tregs and Th17 Cells

While analyzing the difference of surface marker expression in dependency of the severity of TBI, we were able to determine a significant increase in the expression of CD69 on CD4+ Tregs (B = 0.040, SE = 0.0080, 95% CI [0.024, 0.056], *p* <0.001) as seen in Figure 1C, and Th17 cells (B = 0.028, SE = 0.0093, 95% CI [0.010, 0.046], *p* = 0.003) shown in Figure 1D for the non-severe TBI population, whereas the population of severe TBI patients had no significant change in the timespan of 10 days in the expression of CD69 on CD4+ Tregs (B = 0.020, SE = 0.0163, 95% CI [−0.011, 0.052], *p* = 0.202), seen in Figure 1E, and Th17 cells (B = 0.101, SE = 0.0791, 95% CI [−0.054, 0.256], *p* = 0.202) illustrated in Figure 1F.

### 3.4. Increased Platelet Counts

Although platelet activation markers did not change significantly over the 10-day period, there was a significant increase in platelet count (B = 23.190, SE = 3.3534, 95% CI [16.226, 30.155], *p* < 0.001), shown in Figure 2A. Looking at the difference in TBI patients, we see a higher regression (B = 28.443, SE = 6.62600, 95% CI [16.174, 40.713], *p* < 0.001), as seen in Figure 2B, compared to non-severe TBI patients (B = 20.456, SE = 4.2685, 95% CI [12.0290, 28.822], *p* < 0.001), shown in Figure 2C.

### 3.5. Increase in Coagulation Function Independent from Thrombocyte Function

ROTEM^®^ analysis showed signs of hypercoagulation with decreased clot formation time (CFT) and increased maximum clot firmness (MCF). There was an increase in the maximum clot firmness (MCF) in the extem- (B = 2.079, SE = 0.3016, 95% CI [1.488, 2.670], *p* < 0.001), Figure 3A, intem-(B = 2.458, SE = 0.1717, 95% CI [2.121, 2.794], *p* < 0.001), Figure 3B, and fibtem-measurement (B = 3.441, SE = 0.1761, 95% CI [0.072, 0.093], *p* < 0.001), Figure 3C. At the same time, thrombocytic coagulation function–represented by the subtraction of extem MCF and fibtem MCF–decreased over time (B = 1.153, SE = 0.2243, 95% CI [−1.593; −0.714]), shown in Figure 3D. This decrease was significant (*p* <0.001), showing an impaired platelet function. In addition, we found a decrease in clot formation time (CFT) both in extrinsic (B = −9.858, SE = 3.0001, 95% CI [−15.738; −3.987], *p* = 0.001), shown in Figure 3E, and intrinsic coagulation pathways (B = −9.361, SE = 3.2470, 95% CI [−15.725; −2.997], *p* < 0.01), shown in Figure 3F.

### 3.6. Subgroup Analysis for Patients with Traumatic Brain Injury

An overview of our subgroup analysis for patients with TBI is shown in Table 2. In non-severe TBI patients, a significant increase in the expression of CD69 on CD4+ Tregs and on Th17 cells could be detected; whereas, in patients with severe TBI, the expression of CD69 on CD4+ Tregs and Th17 cells did not reach the level of significance. Platelet count increased in both subgroups significantly over the timespan of 10 days, the existence of TBI did not influence posttraumatic platelet expansion.

## 4. Discussion

While trauma-related injuries are the most common cause of death in young people, and part of the late mortality after trauma is due to impaired immune regulation, there are currently no reliable diagnostic or therapeutic methods for dealing with immunological imbalances. Disorders of immune regulation can lead to both SIRS and sepsis, as well as organ damage and multi-organ failure. A better understanding of the mechanisms of these immune reactions should help to improve future diagnostic and therapeutic options for multiple trauma patients.

In this non-interventional prospective study, the immune response of CD4+ Tregs and Th17 cells in a post-traumatic setting in humans with and without severe traumatic brain injury was investigated. Peripheral blood from 30 multiple trauma patients was collected at nine specific time points up to ten days after hospitalization. By flow cytometry and rotational thromboelastometry, the activation of CD4+ Tregs and Th17 cells, as well as hemostasis and platelet function were investigated. In addition, we differentiated the population based on the severity of TBI (severe vs. non-severe) and examined the expression in the individual subgroups separately.

### 4.1. CD69 Expression on CD4+ Tregs and Th17 Cells

In the first ten days after multiple traumas, we could show an increasing CD69 expression on CD4+ Tregs, however not on Th17 cells, suggesting injury-induced differential cell activation. CD69 is mainly detected in chronic inflammation and at sites of active immune responses and seems to not only modulate inflammatory processes but also play an important role in the regulation of CD4+ Tregs and Th17 cells [35,36]. Other studies have already shown the induction and activation of CD4+ Tregs following severe injury [14]. We have earlier been able to show in a murine burn model that platelets interact with CD4+ Tregs; we were able to demonstrate reciprocal activation in this setting [27]. However, the mechanisms underlying CD4+ Treg activation and platelet interaction are still largely unexplained, especially as data obtained in human studies are missing. Furthermore, the exact roles played by the various subtypes of regulatory T cells are still unclear and the subject of current research. We deliberately did not differentiate between different CD4+ Treg subtypes, as influence and interaction with thrombocytes in this post-traumatic setting should generally be addressed. However, these cells show plasticity enabling, amongst other things, the conversion of regulatory T cells in Th17 cells [37].

### 4.2. Platelet Count and Platelet Function

While the number of platelets after multiple trauma increased significantly, there was a decrease in platelet coagulation function. Intravascular coagulopathy, which is not uncommon after severe injury and immobilization, may have even attenuated the effect of increased platelet count. On the other hand, the transfusion of blood products, which has not previously been considered, could be related to the increase in platelet count. The limited function of transfused platelets could have influenced coagulation as well as immunological functions [38]. It is known that platelets interact with cells of the immune system. Many studies describe the importance of platelets for the regulation of immune system function [20]. Although the number of platelets after trauma correlates with mortality and seems to have prognostic significance, the role of platelets after trauma is still under discussion in literature. While platelets have long been known for their coagulation function, an increasing number of studies show the interaction with cells of the immune system and their roles in various immune reactions. They are a component of inflammatory and immune reactions and play an important role in innate and acquired immunity [22,39]. Platelets promote the differentiation of CD4+ T cells, as well as their cytokine production and release, including the cytokines characteristic for Th17 cells and CD4+ Tregs [40]. They form cell–cell contacts with lymphocytes and can stimulate or inhibit their activity as well as support the transendothelial migration of these cells [21]. Platelets have been shown to directly affect both CD4+ Tregs and Th17 cells. While they promote the CD4+ Treg response, they appear to have a biphasic influence on Th17 cells [23]. Since many platelet-derived mediators are very specific, the modulation of platelet activation could help to regulate immune activation, improve pathogen clearance and mitigate tissue damage in severe inflammation with fewer side effects than broad-spectrum immunosuppression [39]. The platelet count and activation could be used in a clinical context as a diagnostic marker for immunological dysbalance as well as for the prognosis of patient outcomes in trauma and sepsis patients. In addition, platelets could be a target for immunomodulatory therapy.

### 4.3. Thromboelastometry

We found an increase in coagulation function in the extrinsic and intrinsic coagulation pathways, expressed in a shorter clot formation time, and a higher clot firmness over time. This increase was mainly due to plasmatic coagulation factors. However, blood transfusions were not included in the evaluation and may have influenced the outcome of the coagulation measurement. Since ROTEM^®^ is an artificial system, influencing factors such as flow behavior, drug effects and interactions with endothelial cells and other molecules cannot be adequately mapped. Nevertheless, many studies conclude that ROTEM^®^, particularly in trauma patients, is well suited for the diagnosis of in vivo coagulation disorders [41]. The influence of trauma on coagulation has long been known and can be observed frequently. In a study by Engelman and coworkers, 85% of multiple-trauma patients showed signs of hypercoagulability [42]. This traumatic coagulopathy is associated with poor clinical outcomes and can thus be of clinical importance [43]. An increasing number of studies show connections between coagulation and immune function. The coagulation system and the inflammatory response seem to influence each other [44]. Systemic inflammation activates coagulation and inhibits anticoagulant mechanisms and fibrinolysis [45]. At the same time, coagulation activation and fibrin deposition help to ward off infectious agents and reduce the inflammatory response [46]. The time course of coagulation functions, as well as responsible mechanisms and their influencing factors, should be further investigated in the future. A better understanding of these processes and their components could contribute to the prognosis, as well as diagnosis and regulation of hemostatic and immunological dysbalances.

### 4.4. Influence of TBI on the Expression of CD69 on CD4+ Tregs and Th17 Cells

Neuroinflammation is currently critically discussed and seems to aggravate secondary injury after traumatic brain injury [47]. On the contrary, moderate acute inflammation after trauma showed a benefit for neurological restoration [48]. We were able to show that patients without severe TBI show a significant increase in expression of CD69 on CD4+ Tregs and Th17 cells, whereas patients with severe TBI reveal no change over the time span of 10 days after trauma. The function and immunological role of CD69 is already described above. Brain injury may contribute less due to activation of CD69 than injuries in other regions.

It was shown that CD4+ Tregs have a beneficial outcome as part of the posttraumatic neuroinflammatory response shortly after TBI [49]. Chronic inflammation following acute trauma may be linked to neurodegeneration and a variety of diseases including Alzheimer’s disease, Parkinson’s disease and amyotrophic lateral sclerosis (ALS) [50]. In addition, in ischemic strokes, emerging evidence has shown that CD4+ Tregs play a critical role in maintaining immune hemostasis and suppression of immune responses [51].

There have been several indications for therapeutic approaches regarding neuroinflammation and the immune response after TBI [52,53,54]. It is a highly-targeted topic and will need further investigation to optimize the treatment of TBI patients.

### 4.5. Increase in Platelet Count for Severe TBI and Non-Severe TBI Patients

After trauma, all patients showed a significant increase in platelet count during the first 10 days. Severe TBI patients had a higher increase than non-severe TBI patients. Elevated levels of platelets after severe TBI were previously described; however, increased platelet count may also be due to multiple injuries or severity of injury [55]. Our population of severe TBI patients showed a slightly higher mean ISS than the non-severe TBI population (37 ± 12 vs. 30 ± 11). An isolated TBI in patients, as well as in a rat model, showed to induce platelet dysfunction [56,57]. Of special importance is the history of the patient. Geriatric patients with TBI are more likely to have comorbidities, which makes medicating with the anticoagulant or antiplatelet medication necessary. The careful anamnesis and subsequent management of hemostasis are important aspects of the treatment of geriatric TBI patients [58].

### 4.6. Limitations

Our study has limitations regarding scope and methodology. As this is a descriptive, non-interventional study, causalities cannot be proven. The patient population was small, comprising only 30 patients, which makes subgroup and correlation analysis difficult. Since we did not conduct any further examinations after the first 10 days after trauma, we cannot conclude on the normalization of T cell activation and platelet count. Furthermore, clinical outcome (e.g., nosocomial infections) was not assessed and considered in this study. In addition, the use of blood products has not been considered. The lack of specific antibodies for Th17 cells and the exclusive use of surface antibodies for the identification of CD4+ Tregs should also be considered. However, studies have shown that the combination of CD4+, CD161+ and CD196+ used for the identification of Th17 cells provides good results and that the combination of antibodies against CD4+, CD25+ and CD127+ represents a useful alternative to the use of the intracellular marker FoxP3 [11,59]. Platelet function can only be determined using aggregometry, our ROTEM data provides a surrogate marker (MCF extem-fibtem), as has been described in literature before [34].

## 5. Conclusions

In conclusion, we describe an increase in CD69 expression on CD4+ Tregs, but not Th17 cells, with a simultaneous increasing number of platelets after severe injury in a clinical study. The increase in platelets with concomitant declining platelet clotting function supports the concept of post-traumatic platelet dysfunction. Coagulation changes indicate coagulopathy, which is known to be related to the need for blood products, organ failure and increased mortality. Furthermore, our results support the concept of injury-specific immune responses, we describe a significantly higher increase in CD69 on CD4+ Tregs and Th17 cells in non-severe TBI patients as compared to severe TBI patients.

CD4+ Tregs are considered key players in the immune response after trauma and seem to be protective, while platelets seem to modulate the immune system after severe injury. The presence of TBI modulates the post-traumatic immune function depending on severity. This study contributes to a better understanding of the interaction of CD4+ Tregs, Th17 cells, and platelets. Our results gain importance in the future development of post-traumatic immunomodulatory therapies, but also in strategies to improve monitoring of the immune system following injury.

## Figures and Tables

**Figure 1 jcm-11-05315-f001:**
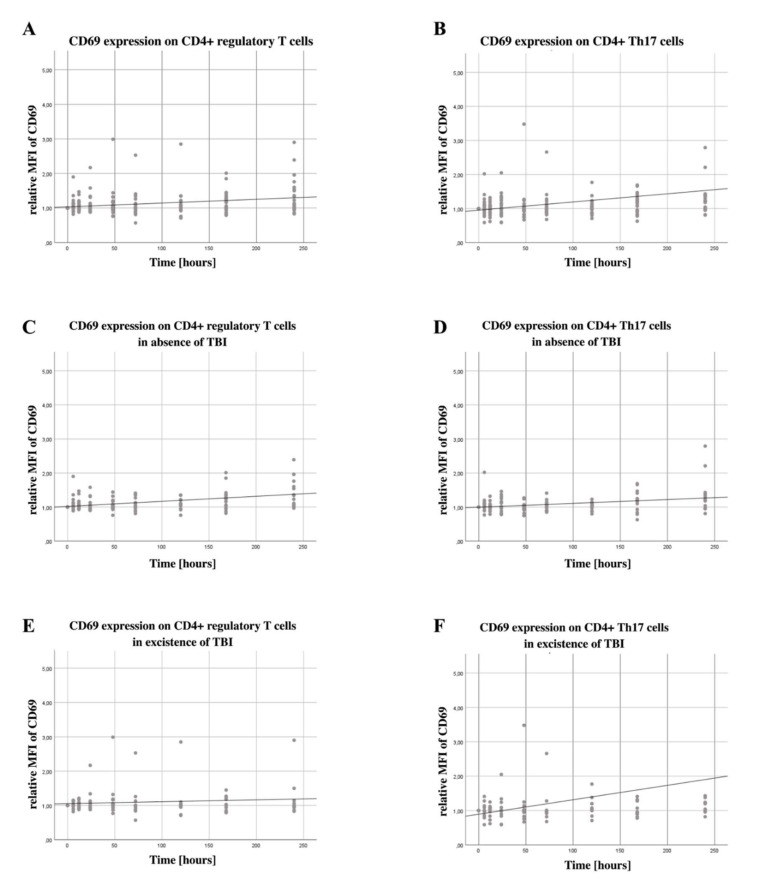
**CD69 expression on CD4+ regulatory T cells and Th 17cells over the first ten days after trauma in patients with and without severe TBI**. (**A**) The CD69 expression on CD4+ Tregs and Th17 cells was measured via flow cytometry using PE-conjugated anti-CD69 antibodies. CD4+ Tregs were defined as CD4+, CD25+ and CD127low. Th17 cells were defined as CD4+, CD161+ and CD196+. Blood was drawn on arrival at the emergency room (0), after 6, 12, 24 and 72 h, as well as after 5, 7 and 10 days. During this time, we found a significant increase in CD69 expression on CD4 Tregs (B = 0.033, SE = 0.0087, 95% CI [0.016, 0.050], *p* < 0.001), (**B**) but not on Th17 cells (B = 0.058, SE = 0.0347, 95% CI [−0.010, −0.0126, *p* = 0.093). (**C**) Looking at the population with absence of severe TBI, we were able to determine a significant increase in the expression of CD69 on CD4+ Tregs (B = 0.040, SE = 0.0080, 95% CI [0.024, 0.056], *p* < 0.001) and (**D**) Th17 cells (B = 0.028, SE = 0.0093, 95% CI [0.010, 0.046], *p* = 0.003). (**E**) The population of severe TBI patients had no significant change in the time span of 10 days in the expression of CD69 on CD4+ Tregs (B = 0.020, SE = 0.0163, 95% CI [−0.011, 0.052], *p* = 0.202) and (**F**) Th17 cells (B = 0.101, SE = 0.0791, 95% CI [−0.054, 0.256], *p* = 0.202). The regression line shown here is for illustrative purposes.

**Figure 2 jcm-11-05315-f002:**
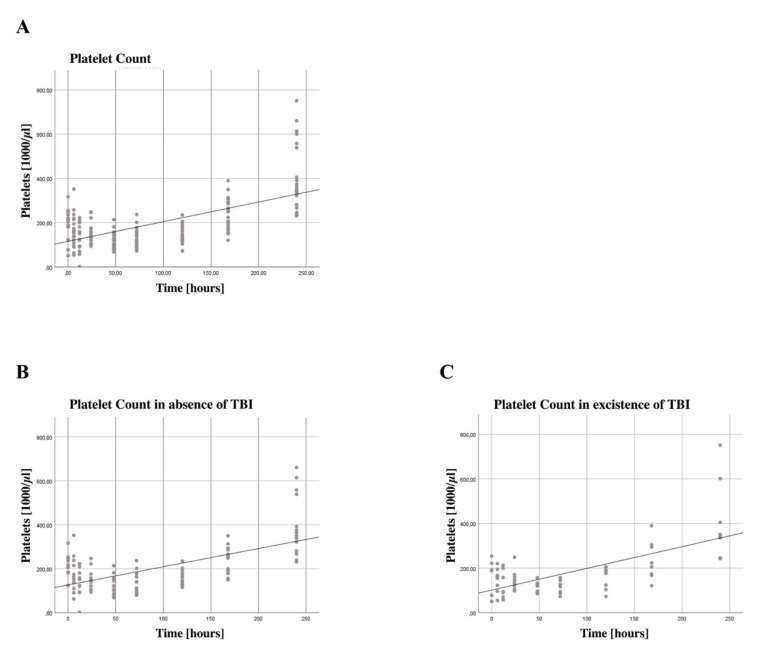
**Absolute platelet count increased significantly over the first ten days after trauma.** (**A**) The absolute platelet count was assessed in the emergency room (0), after 6, 12, 24 and 72 h, as well as after 5, 7 and 10 days. Generalized estimating equations (GEEs) were performed to test for statistically significant changes over time. We found a significant increase in platelet count (B = 23.190, SE = 3.3534, 95% CI [16.226, 30.155], *p* < 0.001), in the first 10 days after trauma for all patients. Patients without a severe injury to the brain (**B**) showed a lower increase (B = 20.456, SE = 4.2685, 95% CI [12.090, 28.822], *p* < 0.001) than patients with a severe TBI (**C**) (B = 28.443, SE = 6.2600, 95% CI [40.713, 20.645], *p* < 0.001). The regression line shown here is for illustrative purposes.

**Figure 3 jcm-11-05315-f003:**
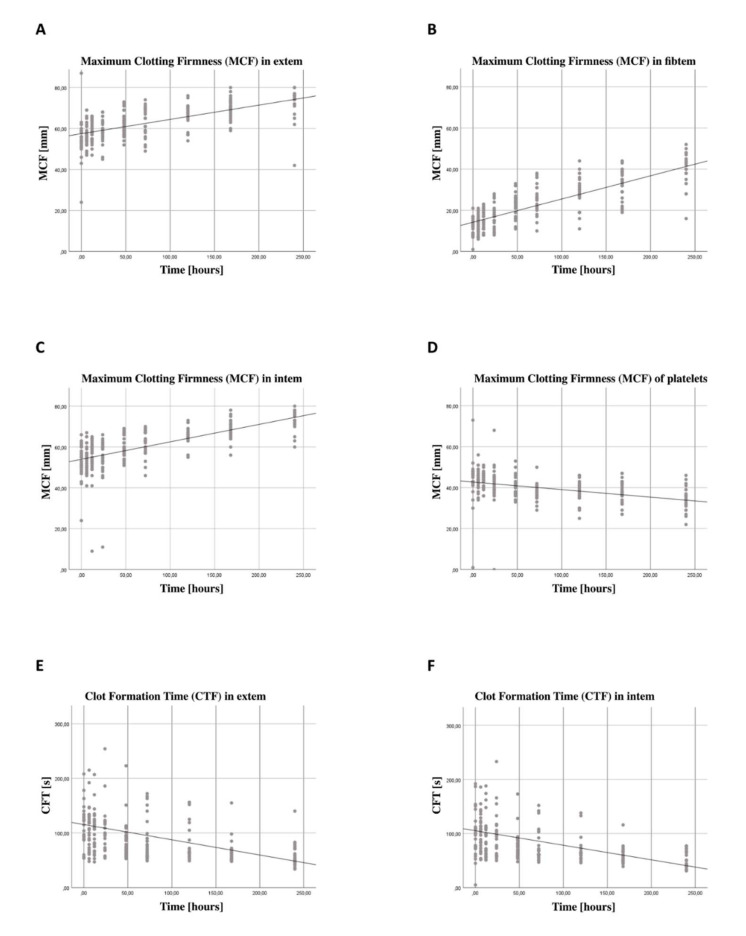
**Thromboelastometric findings revealed a significant increase in MCF in extem, intem and fibtem, while platelet MCF decreased; CFT decreased in extem and intem**. (**A**) The thromboelastometric data were collected using a ROTEM^®^ delta device. Blood was drawn on arrival in the emergency room (0), after 6, 12, 24 and 72 h, as well as after 5, 7 and 10 days. The maximum clot firmness (MCF) on the extem panel, representing the coagulation function in the extrinsic coagulation pathway, increased significantly over the first 10 days after trauma (B = 2.079, SE = 0.3016, 95% CI [1.488, 2.670], *p* < 0.001). (**B**) The same applies to the intem panel, representing the coagulation function in the intrinsic coagulation pathway, (B = 2.458, SE = 0.1717, 95% CI [2.121, 2.794], *p* < 0.001), and (**C**) the fibtem panel, representing the extrinsic coagulation pathway without the contribution of platelets, as these are inhibited by cytochalasin D, (B = 3.441, SE = 0.1761, 95% CI [0.072, 0.093], *p* < 0.001). (**D**) The platelet MCF, however, represented by the subtraction between extem MCF and fibtem MCF, decreased over time (B = −1.153, SE = 0.2243, 95% CI [-1.593; -0.714]), *p* < 0.001), which is indicative of a platelet dysfunction. As the clot gets firmer over time, it also forms faster. (**E**) The clot formation time (CFT) decreases in both extrinsic (B = −9.858, SE = 3.0001, 95% CI [−15.738; −3.987], *p* = 0.001), and (**F**) intrinsic coagulation pathways (B = −9.361, SE = 3.2470, 95% CI [−15.725; −2.997], *p* < 0.01). The regression line shown here is for illustrative purposes.

**Table 1 jcm-11-05315-t001:** Demographic patient data.

Characteristics	Frequency (Percentage)	Mean ± SD
Patient number	30 (100%)	
Age [years]		50.5 ± 18.5
Sex		
male	18 (60%)	
female	12 (40%)	
ISS		33.2 ± 11.5
Survivors	26 (87%)	
ICU stay [days]		13.3 ± 15.3
Number of surgeries		3.9 ± 2.4
AIS score		
Head/neck	20 (67%)	2.6 ± 1.3
Face	15 (50%)	1.7 ± 1.0
Chest	24 (80%)	3.2 ± 1.2
Abdominal/pelvic contents	11 (37%)	2.4 ± 1.6
Extremities/pelvic girdle	27 (90%)	3.0 ± 0.8
External	19 (63%)	1.3 ± 0.7
Injury mechanism		
Road traffic accident	20 (66.7%)	
Suspected suicide attempt	5 (16.7%)	
Others	5 (16.6%)	

**Table 2 jcm-11-05315-t002:** Overview of the subgroup analysis for patients with traumatic brain injury. AIS, Abbreviated Injury Severity Score.

Value	All Patients		Non-Severe TBI Population (AIS Head Region 1–2)		Severe TBI Population (AIS Head Region ≥ 3)	
	B	*p*-Value	B	*p*-Value	B	*p*-Value
CD69 on CD4+ Tregs	0.033	<0.001	0.040	<0.001	0.020	0.202
CD69 on CD4+ Th17 cells	0.058	0.093	0.028	0.003	0.101	0.202
Platelet count	23.190	<0.001	20.456	<0.001	28.443	<0.001

## Data Availability

All data generated or analyzed during this study are included in this published article.

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
