# Peer review of "Traumatic Brain Injury Induces a Differential Immune Response in Polytrauma Patients; Prospective Analysis of CD69 Expression on T Cells and Platelet Expansion"

_jcm, 2022, doi:10.3390/jcm11185315_

Round 1

Reviewer 1 Report (Previous Reviewer 1)

Dear authors,

I am honored to be able to participate in the revised review.

I checked the revised manuscript. The treatise was replaced and the "thymus" part was removed.

However, "Figure (1-3)" has not changed.

Very small and low resolution. You need to come up with a way to publish your treatise and make it easier to understand.

Author Response

Thank you for the suggestion to increase readability and clarity of the figures. We increased the size and the solution of the figures according to your suggestions. Please see the attached finalized document.

Reviewer 2 Report (New Reviewer)

Overall the manuscript is too long and includes a lot of repetition. Each section should be more succinct including the title. I would suggest for the title "Traumatic brain injury induces a differential immune response in polytrauma patients; prospective analysis of CD69 expression on T cells and platelet expansion.

Abstract covers the key concepts and findings of the paper. Line 23 could be improved as follows: Subgroup analysis was performed to assess differences between polytrauma patients with and without severe traumatic brain injury (TBI). Line 24 could be improved as follows: In this non-interventional prospective clinical trial, we analysed sequential blood samples over a period of 10 days from 30 patients after multiple trauma with an ISS >16. 

Introduction needs significant refinement and possible restructuring. It is not entirely clear whether the subgroup analysis of severe TBI patients was a primary aim. If so, this concept would be better introduced earlier in the Introduction. Some of the detail in the TBI section should be removed particularly clinical variables that are not measured in the study, e.g., VAP, ED and ICU management, post traumatic cerebral infarction. Too many concepts confuse the reader and distract from the The section on CD4+ in particular is too long providing a detailed description of the development and the transcription factor FoxP3 which do not add value to the manuscript. The Introduction is not the place to present the study's findings (Line 144-152). The Introduction should finish with the aim of the study.

Methods: Some detail is missing on patient selection. Were all consecutive eligible patients included? This seems unlikely since only 30 patients were enrolled over 4 years. Specify whether inclusion criteria was multiple-injured patients only (no isolated injury). There is a lot of repetition in the Methods, e.g. Line 198 Fc-block. The section "Experimental Protocols" could be removed with the detail included in the appropriate method section (flow cytometry or thromboelastometry). What analyser was used for platelet count measurements? No detail is provided on the approximate time from injury the first blood sample was taken. Variability in time from injury to arriving at hospital may have significantly influenced results and has not been mentioned (even as a potential limitation). Technically thromboelastometry is not an indicator of platelet activation (see comments on Results below). Line 221 on the history of TEM is unnecessary. HEP-tem test may have been more accurate than IN-tem to remove the effect of any heparin (especially in later stages when patients may have received LMWH to reduce thrombosis). Statistics - was there any assessment of data normality to determine the appropriate data presentation and analysis? If the subgroup analysis of severe TBI patients was determined a priori this should be outlined in the Methods. The methods also don't specify how missing data was handled, e.g. in non-survivors who did not have all representative blood samples over the 10 day monitoring period.

Results - The text on the study population does not align with what is presented in Table 1. The age, % male and female, % survivors, number of surgeries and ICU days are all inconsistent between the text and the Table. The dot points should be removed from the Table. Line 271 - remove "we were pleased" - results should be written as found without emotion. The vertical scales on the Figures should be reconsidered. The visual representation in its current form does not give the reader an easy sense of change or differences. Extem MCF - fibtem MCF is only a surrogate marker and not a true indicator of platelet function. Platelet function can only be determined using aggregometry. A reduced Extem MCF - fibtem MCF may also be due to increased fibrinogen, the production of which is unregulated in the liver after trauma, and is also commonly given to patients as fibrinogen concentrate. Table 2 repeats results that have already been presented in the previous Figures.

Discussion: Line 365 states that this is a pilot study however that terminology was not used in the previous sections. Some parts of the Discussion repeat the Introduction or include more Background information that is not required for the message of the paper, e.g. Line 376-378, Line 385-388. The term plasmatic coagulation is generally used for plasma-based coagulation tests (e.g. PT, aPTT), not ROTEM which assesses whole blood. Line 421 talks about hyperfibrinolysis however hyperfibrinolysis was not assessed/measured (e.g. by FIBTEM maximum lysis or lysis index).Line 432 - the study didn't actually assess the interaction between CD4+ Tregs and platelets - this would require correlation analysis. Line 433 was already introduced under the Subheading CD69 expression of CD4+ T regs and Th17 cells. Line 455 - The propensity of evidence shows that neuroinflammation contributes to secondary injury after TBI. The Discussion has insufficient critical evaluation and relies too much on the conclusion that further investigation is required.

References - of the 65 references cited only 2 are from the previous 3 years. This is a field of research that has been growing exponentially and more up-to-date literature should be included.

Author Response

Overall the manuscript is too long and includes a lot of repetition. Each section should be more succinct including the title. I would suggest for the title "Traumatic brain injury induces a differential immune response in polytrauma patients; prospective analysis of CD69 expression on T cells and platelet expansion.

Thank you for the valuable suggestion. The overall manuscript was shortened and revision was undertaken to make it more succinct. We changed the title of the manuscript accordingly.

Abstract covers the key concepts and findings of the paper. Line 23 could be improved as follows: Subgroup analysis was performed to assess differences between polytrauma patients with and without severe traumatic brain injury (TBI). Line 24 could be improved as follows: In this non-interventional prospective clinical trial, we analysed sequential blood samples over a period of 10 days from 30 patients after multiple trauma with an ISS >16. 

Thank you for the valuable suggestions and comments. We changed the abstract accordingly to the suggestions of the reviewer.

Introduction needs significant refinement and possible restructuring. It is not entirely clear whether the subgroup analysis of severe TBI patients was a primary aim. If so, this concept would be better introduced earlier in the Introduction. Some of the detail in the TBI section should be removed particularly clinical variables that are not measured in the study, e.g., VAP, ED and ICU management, post traumatic cerebral infarction. Too many concepts confuse the reader and distract from the The section on CD4+ in particular is too long providing a detailed description of the development and the transcription factor FoxP3 which do not add value to the manuscript. The Introduction is not the place to present the study's findings (Line 144-152). The Introduction should finish with the aim of the study.

Thank you for the valuable suggestions and comments. We revised the introduction accordingly to make it more concise and clearer to understand. Especially, we shortened the sections on CD4+ Tregs and TBI in order to avoid confusion of the reader. The subgroup analysis of severe TBI was not primary aim of the study, now made clear in the revised introduction. The last paragraph of the introduction was significantly shortened, the study’s findings were removed and the aim of the study was highlighted.

Methods: Some detail is missing on patient selection. Were all consecutive eligible patients included? This seems unlikely since only 30 patients were enrolled over 4 years. Specify whether inclusion criteria was multiple-injured patients only (no isolated injury). There is a lot of repetition in the Methods, e.g. Line 198 Fc-block. The section "Experimental Protocols" could be removed with the detail included in the appropriate method section (flow cytometry or thromboelastometry). What analyser was used for platelet count measurements? No detail is provided on the approximate time from injury the first blood sample was taken. Variability in time from injury to arriving at hospital may have significantly influenced results and has not been mentioned (even as a potential limitation). Technically thromboelastometry is not an indicator of platelet activation (see comments on Results below). Line 221 on the history of TEM is unnecessary. HEP-tem test may have been more accurate than IN-tem to remove the effect of any heparin (especially in later stages when patients may have received LMWH to reduce thrombosis). Statistics - was there any assessment of data normality to determine the appropriate data presentation and analysis? If the subgroup analysis of severe TBI patients was determined a priori this should be outlined in the Methods. The methods also don't specify how missing data was handled, e.g. in non-survivors who did not have all representative blood samples over the 10 day monitoring period.

Thank you for the valuable suggestions and comments. We revised the methods section accordingly. Details on patient inclusion were added. Patients were included only when the admission was no more than 12h after injury (no referrals from other hospitals). Furthermore, inclusion of the patient was only conducted when analysis could be guaranteed for the full follow-up period (availability of the study team). Furthermore, isolated injury was excluded. We added these details to the methods section. Repetitions were deleted (e.g. Fc block), additionally we removed the section ‘experimental protocols’ to avoid redundancy. Platelet count was assessed in the routine blood work, this was added. The mean time from injury to admission in the trauma bay was 60.5 min in the study period, we added this piece of information to the blood sample/ data retrieval section. We agree, that ROTEM analysis only provides a surrogate marker for platelet function (MCF extem-fibtem). We added this piece of information including referring to the literature (Lang et al., 2006) to the methods section and the limitations section. We agree, that HEP-tem would have provided interesting further information, due to the logistics of the present study (sequential blood samples) we had to limit the number of tests conducted.

Concerning statistics we would like to point out, that GEEs were chosen due to the significant advantages in longitudinal studies. The GEE method was developed by Liang and Zeger (1986) in order to produce regression estimates when analyzing repeated measures with non-normal response variables. GEE can take into account the correlation of within-subject data (longitudinal studies) and other studies in which data are clustered within subgroups. Failure to take into account correlation would lead to the regression estimates (Bs) being less efficient- meaning they would be more widely scattered around the true population value. An assessment of normality is not needed. Missing data is in most longitudinal studies frequent, the strength of GEEs is to manage missing data without loss in efficiency.

As mentioned, GEE can take into account the correlation of within-subject data (longitudinal studies) and other studies in which data are clustered within subgroups. The method is therefore highly suitable for TBI subgroup analysis in our study.

Results - The text on the study population does not align with what is presented in Table 1. The age, % male and female, % survivors, number of surgeries and ICU days are all inconsistent between the text and the Table. The dot points should be removed from the Table.Line 271 - remove "we were pleased" - results should be written as found without emotion. The vertical scales on the Figures should be reconsidered. The visual representation in its current form does not give the reader an easy sense of change or differences. Extem MCF - fibtem MCF is only a surrogate marker and not a true indicator of platelet function. Platelet function can only be determined using aggregometry. A reduced Extem MCF - fibtem MCF may also be due to increased fibrinogen, the production of which is unregulated in the liver after trauma, and is also commonly given to patients as fibrinogen concentrate. Table 2 repeats results that have already been presented in the previous Figures.

Thank you for the valuable suggestions and comments. We apologize for the confusion between the table and the text. The inconsistent data was corrected. Furthermore, we removed the dot points in the table, as suggested. We removed the phrasing ‘we were pleased’ in line 271. We increased the size of the figures, allowing for better visualization. Furthermore, the resolution of the figures was increased. We deliberately chose to apply the same vertical scale for the adjacent panels in one figure in order to facilitate easier identification of changes or differences between the parameters / subgroups.

Concerning ROTEM analysis, please see above. We applied changes to the methods section and the limitations.

Concerning Table 2 we agree that a certain degree of repetition is given. Nevertheless, the table provides highly interesting data, which is not given in the panels of the figure. By addition of the table the data of the subgroup analysis is underlined, the reader is provided with the B and the p-value of the GEE analysis.

Discussion: Line 365 states that this is a pilot study however that terminology was not used in the previous sections. Some parts of the Discussion repeat the Introduction or include more Background information that is not required for the message of the paper, e.g. Line 376-378, Line 385-388. The term plasmatic coagulation is generally used for plasma-based coagulation tests (e.g. PT, aPTT), not ROTEM which assesses whole blood. Line 421 talks about hyperfibrinolysis however hyperfibrinolysis was not assessed/measured (e.g. by FIBTEM maximum lysis or lysis index).Line 432 - the study didn't actually assess the interaction between CD4+ Tregs and platelets - this would require correlation analysis. Line 433 was already introduced under the Subheading CD69 expression of CD4+ T regs and Th17 cells. Line 455 - The propensity of evidence shows that neuroinflammation contributes to secondary injury after TBI. The Discussion has insufficient critical evaluation and relies too much on the conclusion that further investigation is required.

References - of the 65 references cited only 2 are from the previous 3 years. This is a field of research that has been growing exponentially and more up-to-date literature should be included.

Thank you for the valuable suggestions and comments. We deleted the term pilot study. Furthermore, we deleted redundant parts of the discussion as suggested. The term plasmatic coagulation was deleted and changed into thrombelastometry. The sentence concerning hyperfibrinolysis was deleted as suggested. We agree, that no correlation analysis was conducted to assess interaction between Tregs and platelets. We deleted the paragraph and included information/ discussion on platelet function in the above named section platelet function. Line 433 was deleted as suggested. A more pronounced discussion and introduction of newest literature was added for neuroinflammation. We agree and underline in our discussion and conclusion, that our data provides ground for further investigation. We underline the strength of our study, being the first prospective clinical trial with sequential blood draws investigating immunologic differences (adaptive immune system and platelets).

Finally, we added newest literature and increased the ratio of more up-to- date literature.

Round 2

Reviewer 2 Report (New Reviewer)

The authors have addressed most of the previous comments and improved the manuscript. Please note it is always much easier for the reviewer to see how the previous suggestions have been addressed, if changes in the manuscript are highlighted/underlined.

Line 43: Amend sentence as follows: MODS is a serious condition requiring intensive care treatment, yet often and associated with mortality of 30 to 80 percent.

Line 67: Amend sentence as follows: Different subgroups of CD4+ Tregs have been shown, but their exact roles are not yet clear and therefore they are the subject of the current research.

Line 72: Amend sentence as follows: Recent studies conducted by our group described the kinetics and and localisation of CD4+ Treg activation following trauma in a murine model, and found that injury induces rapid activation of CD4+ Tregs, but not CD4+ non-Tregs.

Line 164: Remove "in the study period".

Line 191: Add the version of FlowJo software used.

Statistics: Specify the p-value considered statistically significant, e.g. A p-value of <0.05 was considered statistically significant.

Line 237: Instead of were operated 3.9 +/- 2.4 times, this would be better phrased as "had an average of 3.9 +/- 2.4 surgical procedures.

Line 241-246 - Font and line spacing require correction for consistency with rest of manuscript.

Table 1: Units are required for age.

Line 250: To avoid sounding like repetition of the methods remove the first 2 sentences and just start by presenting the result.

Line 339: Change "in the" to "of", i.e. Disorders of immune regulation...

Line 449: Amend as follows: The patient population was small, comprising only 30 patients, which makes subgroup and correlation analyses difficult.

Line 466: Should be described as a clinical "study" rather than a "clinical trial" as it was not interventional.

Author Response

We would like to thank the reviewer for the revision comments. We changed the manuscript accordingly, all suggested points were carefully addressed.

Please find the word document with yellow highlighted changes attached.

This manuscript is a resubmission of an earlier submission. The following is a list of the peer review reports and author responses from that submission.

Round 1

Reviewer 1 Report

Dear authors,

Thank you for the very interesting manuscript. I think it's an area that so many people are interested in. The dissertation is almost complete, but here are some points I noticed.

1. The first line(line 36) of the introduction, but the literature is old. Please present a little newer literature as it is a mortality rate.

2. "Like all T cell ~"(Line 59)

Is this the same result for humans? If so, the thymus will almost disappear in old age, so it is necessary to analyze by age.

3.It's Figure 1 and 2, but it doesn't look good. Can you make it a little easier to see?

Reviewer 2 Report

Dear editor,

Dear authors

Thank you for the opportunity to review this interesting manuscript. It is a well-written paper, however, I would like to suggest that the following points be addressed to improve the quality of the manuscript.

Introduction:

- A little too long

- Objectives of the study have been described

Methods

- The methods are complete

Results:

- extensive description of the results and presentation of data